# Look but Don't Touch: Gradient Informed Selection Training

## Abstract

The amount of data available for training foundation models is far greater than our amount of compute. In many domains, this will likely always be the case. Further, not all data is equally valuable for learning, and the learning value of data changes over the course of training. To optimize learning in this setting, several active data selection methods have been proposed; however, they either incur significant additional computational costs or offer limited performance benefits. We propose Gradient Informed Selection Training (GIST), an active data selection method that selects a core subset of examples from mini-batches based on their gradient alignment with a small, fixed holdout set taken from the training set. At each training step, GIST computes per-example gradients and selects only those that are most aligned with the holdout gradient, thereby guiding model updates toward better generalization. On the large, noisy web-scraped image dataset Clothing-1M, GIST trains in 3x faster wall clock time, using 6x fewer steps, and achieves 4% higher final accuracy than RHO-LOSS and uniform data selection. We demonstrate the robustness of the method in both the text and image domain.

## 1 Introduction

Modern machine learning models are growing at an unprecedented rate in both size and capability. Foundation models such as CLIP Radford et al. (2021) and GPT Brown et al. (2020) demonstrate that scaling model size and dataset size can significantly improve generalization across diverse tasks. In particular, we focus on vision model training, where the signal-to-noise ratio in data is often lower than in text or structured domains. Real-world vision datasets, such as those collected for applications like autonomous driving, frequently suffer from label noise, occlusions, near-duplicate frames, and other imperfections that are harder to clean at scale Liu et al. (2024); Yun et al. (2021); Idrissi et al. (2022). These challenges make vision tasks especially sensitive to the quality and selection of training data.

Large models can sometimes compensate by learning from vast amounts of data, but this strategy comes at substantial computational cost and can slow convergence or degrade final performance. Selectively focusing on higher-quality or more informative examples offers a promising way to address this inefficiency.

It is well understood that not all training examples contribute equally to learning. Techniques such as curriculum learning, where examples are presented in a progression from easy to difficult Bengio et al. (2009), emphasize the role of the order and quality of training data. However, in practice, the improvements achievable through curriculum learning are limited, both in terms of final accuracy and the practical difficulty of properly ranking samples and pacing their introduction during training Soviany et al. (2022).

This motivates a shift toward active data selection, which offers a more adaptive approach to prioritizing examples. Recent methods such as RHO-LOSS Mindermann et al. (2022) and InfoBatch Qin et al. (2024) have shown that focusing on valuable examples during training can accelerate convergence and improve generalization, particularly in noisy settings. However, we hypothesize that current methods still fall short of realizing the full potential of active selection. Our goal is to push closer to this upper bound and quantitatively demonstrate that a better "curriculum" exists—specifically, that there is a more effective way to prioritize data to enhance model learning.

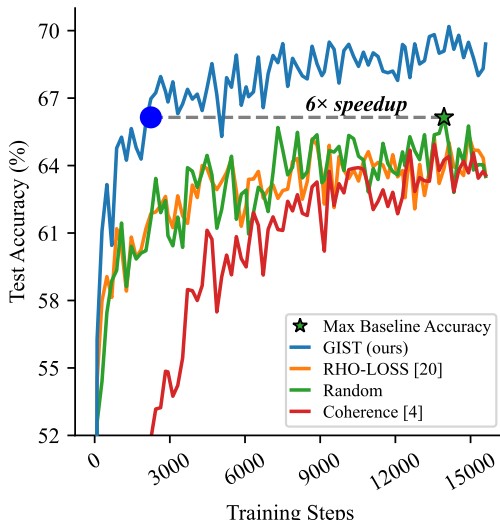

Figure 1: **Acceleration in training for large-scale noisy datasets (Clothing-1M)**. GIST trains in 6x fewer steps and achieves at least 4% accuracy gain across all other baselines (RHO-LOSS, random selection, gradient coherence).

In this work, we provide the following contributions:

- We propose an active data selection approach, **Gradient Informed Selection Training (GIST)**, designed to identify and prioritize the most valuable training examples based on gradient information. Through experiments on noisy datasets such as Clothing1M Xiao et al. (2015), we show that even in the presence of significant label noise, a better training curriculum exists. Our method outperforms prior state-of-the-art methods (i.e. RHO-LOSS), demonstrating that a substantial performance gap still remains between current selection strategies and the optimal subset selection.

- We analyze the performance of data selection methods across a range of **selection fractions**, defined as the proportion of examples chosen from each large batch to form the smaller training batch. We find that GIST outperforms RHO-LOSS at all selection fractions greater than 0.3. At 0.6 GIST trains in 3x faster wall clock time, using 6x fewer steps, and reaches 4% higher final accuracy.

## 2 RELATED WORK

### 2.1 DATA SELECTION APPROACHES

Curriculum learning is one of the earliest structured data selection strategies. First introduced by Bengio et al. Bengio et al. (2009), it involves presenting training examples in a progression from easy to difficult, mimicking the way humans learn. This staged exposure has been shown to accelerate convergence, improve generalization, and enhance model robustness across various domains including vision and language Zhou et al. (2024); Wu et al. (2024); Nguyen et al. (2024); Zhang et al. (2024); Joaquin et al. (2024).

Data subset selection refers to methods aimed at identifying smaller, representative subsets of a larger dataset to reduce computational cost while retaining performance. It differs from data cleaning, which primarily removes or corrects erroneous and mislabeled samples Song et al. (2019); Northcutt et al. (2022) and active data selection, which dynamically adjusts the training data based on feedback during training. Subset selection emphasizes the representativeness of the chosen samples, often trading off exploration (diversity of samples) against exploitation (selecting samples closely aligned with the current model's needs), a balance extensively studied within reinforcement learning paradigms Sutton & Barto (2018).

There have been significant advances in data subset selection methods Coleman et al. (2020); Wei et al. (2014); Kaushal et al. (2019); Mirzasoleiman et al. (2020); Jain et al. (2023). More recently, research has shifted toward active data selection, where the training set is dynamically filtered or prioritized based on model feedback Mindermann et al. (2022); Qin et al. (2024); Killamsetty et al. (2021b); Hacohen & Weinshall (2023); Xie et al. (2023); Deng et al. (2023). The Reducible Holdout Loss (RHO-LOSS), proposed by Mindermann et al. Mindermann et al. (2022), prioritizes examples with high reducible loss: those whose errors can still be improved with further training. This is done through a teacher-student framework: the teacher estimates the irreducible loss of each sample, while the student updates on the most promising ones. RHO-LOSS has been shown to increase training efficiency and improve generalization, and has been extended to large-scale multimodal training in recent work Tschannen et al. (2025). In this work, we propose a simpler, one-network method for active data selection as opposed to Rho-LOSS's two model teacher-student framework.

## 2.2 Gradient-Based Methods

Gradient-based data selection approaches aim to retain only those examples whose gradient contributions are most representative or beneficial Killamsetty et al. (2021a; 2022). Grad-Match, introduced by Killamsetty et al., selects subsets of training data by minimizing the difference between the full-batch gradient and that computed on a subset Killamsetty et al. (2021a). The method identifies a small group of samples whose average gradient closely matches that of the complete dataset, enabling reduced computational overhead while maintaining strong performance.

Gradient coherence, proposed by Chatterjee and Zielinski Chatterjee (2020); Chatterjee & Zielinski (2022), measures the alignment between gradients of individual samples during training. High coherence means that examples push the model in similar directions, which should more strongly support generalization; low coherence, by contrast, signals conflicting dataset features leading to overfitting. In this work, we also experiment with using coherence to actively select examples that collectively steer the model towards having robust learning dynamics during training.

## 3 Gradient Informed Selection Training (GIST)

The success of methods such as RHO-LOSS Mindermann et al. (2022) demonstrates prioritizing training examples expected to yield the highest learning gain can significantly improve model performance, especially in noisy or complex datasets. Yet, current strategies fall short of the performance upper bound achievable if we could always train on the most beneficial examples. This work asks: can we design a method that more effectively identifies high-value training samples to move closer to this ideal?

To approach the theoretical ideal of training on the most informative examples, we propose using a small, fixed holdout set drawn from the training distribution as a stable proxy for the test set. Our method selects training examples whose gradients are best aligned with the average gradient from this holdout set. This alignment encourages model updates that generalize better, without relying on any information from the test set.

Figure 2 provides visual intuition: rather than relying on test set gradients (which are unavailable during training), GIST selects training examples based on how closely their gradients align with the direction of a holdout set gradient. Because the holdout set is never directly used for updates, it encourages generalization without overfitting. By curating this holdout set (e.g., selecting clean or non-ambiguous examples) we can further steer learning toward robustness. While this may risk excluding some long-tail but correct datapoints, we find in practice that the trade-off improves both training efficiency and resilience to label noise. Additionally, we observe that using only last-layer gradients for similarity comparison retains model performance while significantly reducing computational overhead. See Algorithm 1 for the formal procedure and Figure 3 for a visual overview.

## 4 Experiments

We conducted our main experimental evaluation on the Clothing-1 Million dataset, chosen for its noisy real-world challenges. It is the most widely accepted benchmark for image recognition with noisy labels Algan & Ulusoy (2021). To demonstrate the robustness of GIST, we also evaluated

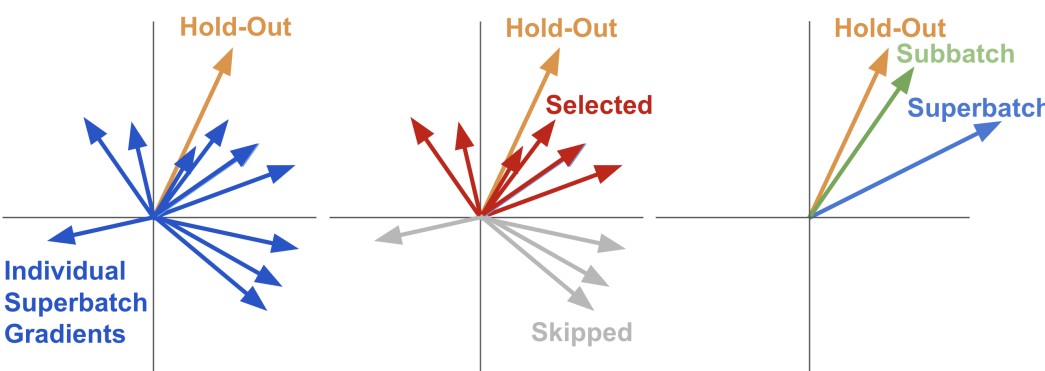

Figure 2: Visual intuition for GIST. Each blue arrow represents the gradient direction, projected into 2 dimensions, for individual training examples within a large batch (**superbatch**). We expect the resultant selected **subbatch** average gradient (represented by the green arrow) of training examples to better approximate the ideal gradient, compared to the gradient computed across the entire super-batch.

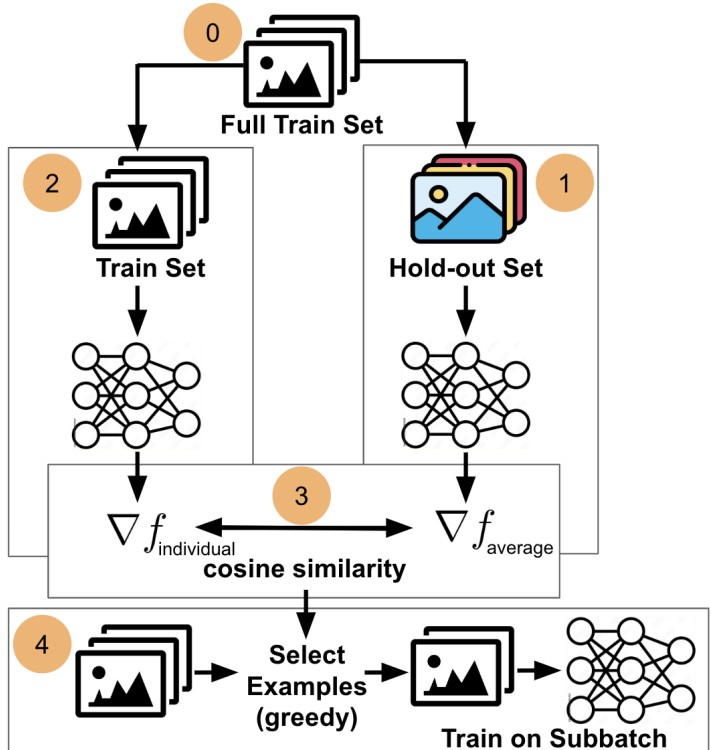

Figure 3: Overview of GIST.

---

**Algorithm 1** Gradient Informed Selection Training (GIST)

---

**Require:** Small holdout set $\mathcal{D}_{\text{ho}}$ (class-balanced, $\ll$ total training size), batch size $n_b$, superbatch
     size $n_B > n_b$, learning rate $\eta$
1: Initialize model parameters $\theta^0$ and time step $t = 0$
2: **for** $t = 0, 1, 2, \ldots$ **do**
3:     Sample a minibatch $\mathcal{D}_{\text{ho}}^{(t)} \subset \mathcal{D}_{\text{ho}}$
4:     HOLDOUTGRAD $\leftarrow$ last-layer gradient of $L(\mathcal{D}_{\text{ho}}^{(t)}; \theta^t)$
5:     Randomly sample a superbatch $\mathcal{B}_t$ of size $n_B$
6:     **for** each $(x_i, y_i)$ in $\mathcal{B}_t$ **do**
7:         TRAINGRAD$[i] \leftarrow$ last-layer gradient of $L(y_i \mid x_i; \theta^t)$
8:         SIMILARITY$[i] \leftarrow \cos\big($TRAINGRAD$[i]$, HOLDOUTGRAD$\big)$
9:     **end for**
10:    $b_t \leftarrow$ top-$n_b$ examples in $\mathcal{B}_t$ with highest SIMILARITY
11:    $g_t \leftarrow$ minibatch gradient on $b_t$ using parameters $\theta^t$
12:    $\theta^{t+1} \leftarrow \theta^t - \eta g_t$
13: **end for**

---

Image Classification on the WebVision dataset Li et al. (2017). In addition, we also demonstrated it's effectiveness in the text modality by training a GPT2 autoregressive text generation model for next token prediction on the OpenWebText2 dataset Brown et al. (2020). Aside from the selection ratio ablations, we used a fixed 0.6 selection ratio for the Image experiments and a fixed 0.375 selection ratio for the text experiments. Additional experiment details can be found in Appendix A.

## 4.1 EXPERIMENT BASELINES

We choose the following baselines to benchmark against:

- **Random:** A simple baseline where subbatches are selected uniformly at random from each superbatch. This serves as a lower-bound reference for selection-based methods.

- **RHO-LOSS:** A teacher-student framework that prioritizes examples with high reducible loss—those for which the model is still expected to improve Mindermann et al. (2022). It has shown state-of-the-art results in noisy settings at low selection fractions, making it a strong baseline on Clothing1M.

- **Coherence:** A gradient-based ablation of GIST that removes the holdout set and instead aligns individual gradients with the average gradient of the current superbatch. This method selects examples whose gradients have the highest cosine similarity to the superbatch gradient, thereby prioritizing coherence within the current training batch rather than generalization to an external validation set.

## 4.2 RESULTS

With the GIST method, we find that we are able to both significantly improve the optimal accuracy/validation loss at convergence and achieve a 2x-6x speedup in terms of both training steps and wall clock time for achieving the same accuracy as the vanilla approach in the 2 modalities of text and image on the Clothing-1M, WebVision, and OpenWebText2. We find that there is an approximately 30% slower steps/sec due to the overhead of calculating truncated per-element gradients for GIST during training, but due to the faster convergence and better results, we were still able to achieve very significant wall clock speedups on these three datasets.

### 4.2.1 ACCURACY IMPROVEMENTS

Using GIST, we were able to achieve significant improvements to the converged performance of our models compared to the vanilla training and all other baseline methods on most of the datasets. We find that all of the other baselines achieved worse or just comparable final performance compared to the vanilla training method. As shown in Figure 4, on Clothing-1M, our method achieved a 4% higher final accuracy. On the WebVision dataset with results shown in Figure 5, we achieved a 5.32%

higher max accuracy with the same number of steps. In addition, even on the same time constraint to account for the additional gradient computation, we achieved a 4.91% higher max accuracy. For the OpenWebText2 dataset, we evaluated our model performance using validation loss with results shown in Figure 6. We achieved 0.051 better validation loss with 100,000 steps compared to vanilla and 0.043 better for the same wall-clock time.

### 4.2.2 TRAINING SPEEDUP

We also demonstrate that GIST is able to achieve significant speedup in training speed over the vanilla training and even the other baselines. As shown in Figure 1 and 4, we are able to achieve a 6x speedup in steps or 3x speedup in wall clock time compared to vanilla training and RHO-LOSS on Clothing-1M. As shown in Figure 5, we are able to achieve a 4.0x iteration speedup and 2.9x speedup in wall clock time. As shown in Figure 6, we are able ot achieve a 4.0x iteration speedup and 2.6x speedup in wall clock time.

### 4.3 SELECTION FRACTION ABLATION

We define the **selection fraction** as the proportion of data selected from the superbatch to form the subbatch used for training. We evaluate all selection fractions from 0.10 to 1.0, in increments of 0.10. Results are shown in Figure 1 for a selection fraction of 0.60. We verify that accuracy gains outweigh additional computational overhead from GIST in Figure 4. We also report the accuracies and speedup for each of these methods at all selection fractions in Figure 8. GIST outperforms all baselines across all selection fractions greater than 0.3, demonstrating its effectiveness in identifying high-value training examples. Meanwhile, RHO-LOSS only outperforms at selection fractions 0.1 and 0.2. The selection fraction analysis for OpenWebText2 is shown in Figure 7, demonstrating that GIST consistently achieves lower validation loss across all tested selection fractions, with optimal performance around 37.5%.

### 4.3.1 ABLATION OF USE OF HELD-OUT SET FOR GRADIENT ALIGNMENT

We also did an ablation of the importance of have a seperate hold-out set which we use to calcuate the gradient alignment. In Figure 5, the "Hold-in" model is a model where we just used randomly selected sections of the train set in each batch to calculate the gradient alignment instead of having a hold-out set which is never trained on. Interestingly, it converges and achieves higher accuracy than the Vanilla training, but significantly worse than the full GIST method.

### 4.4 ANALYSIS OF CLASS SELECTION LIKELY OVER TRAINING EPOCHS

In order to verify and better understand the effect of the GIST selection we examined on a per-class basis how the likelihood of selection changed over time in Figure 9 during training on the Clothing-1M dataset. Interestingly, we found an extremely varied and changing class selection likelyhoods. Some tiny classes like "Sweater" which in the base dataset is only 2.0% compared to an expected class-balanced percent of 7.1% were consistently oversampled during training with relatively little change over the epochs. Other classes like "Jacket" dramatically increased in selection likelyhood as training progressed. "Knitwear", on the otherhand with a relatively similar base proportion to "Jacket" (8.1% vs 8.6%) greatly decreased in sampling likelyhood as training progressed. At the final iteration, "Sweater" was 2.4x more likely to be sampled than "Jacket". We believe these large changes to the selection over the training process demonstrates that our method benefits from this ability to change our selection criteria as the model converges compared to some other data selection methods which build a static set of good or interesting examples. In addition, we also analyzed the set of images which were almost always selected, those which were almost never selected. Some randomly selected examples were are visualized in Appendix A.4. Examples from the smallest classes "Shawl" and "Sweater" appear often in the examples always selected. It appears that the method is automatically doing some oversampling and correction for these small classes. Some of the ones that were only sampled once appear to maybe be some of the hardest or unclear examples such as hoodies where the hood is not visible. We also looked at examples which were never selected early, but always selected late, and those which were selected late, but never selected early. Some examples are show in Appendix A.5. We believe this pattern might correlate to examples that were difficult for the model to learn and examples that were very easy to learn.

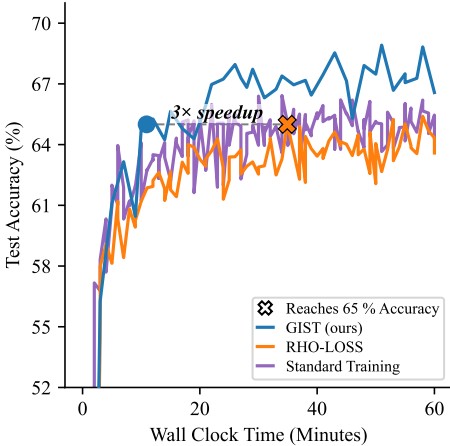

Figure 4: **Comparison of classification accuracy versus training compute cost (clock time) on Clothing-1M at a 0.60 selection fraction.** GIST achieves higher accuracy than baseline methods RHO-LOSS and Standard Training (batch size = selected subbatch size of other methods) when computational cost is held constant across methods.

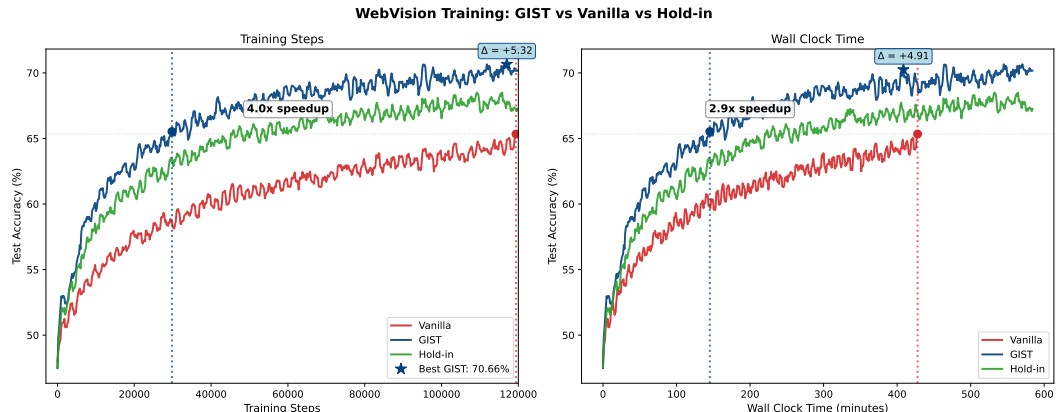

Figure 5: **Acceleration in training for on WebVision**. GIST trains in 4.0x fewer steps, 2.9x faster wall clock, and achieves at least 4.9% better accuracy compared to the baseline for the same wall clock

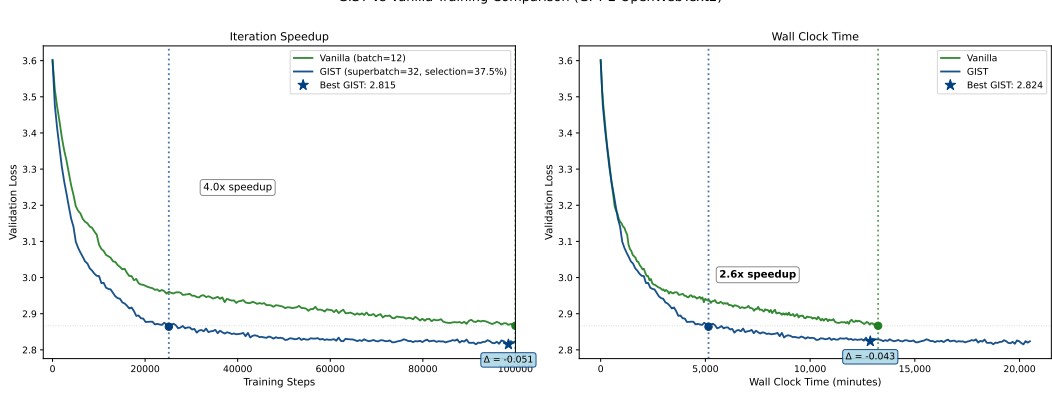

Figure 6: **Acceleration in training for on OpenWebText2**. GIST trains in 4.0x fewer steps, 2.6x faster wall clock, and achieves at least 0.043 better validation loss compared to the baseline for the same wall clock

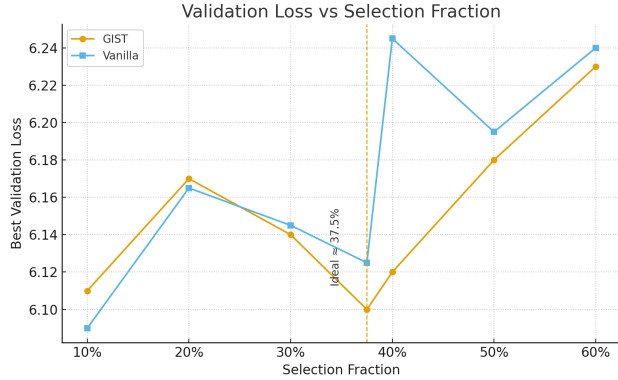

Figure 7: **Validation Loss vs Selection Fraction on OpenWebText2**. GIST consistently achieves lower validation loss compared to vanilla training across all tested selection fractions from 10% to 60%. The optimal selection fraction is around 37.5%, where GIST shows maximum improvement over baseline.

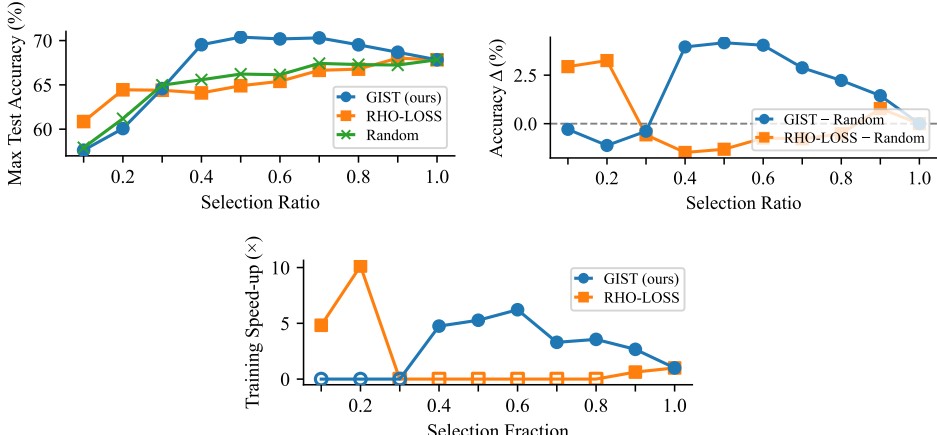

Figure 8: **Top Left:** Max test accuracy achieved across selection fractions; **Top Right:** Difference in accuracy between GIST/RHO-LOSS and random; **Bottom:** Training speedup across selection fractions. Computed by dividing the step at which random selection achieves max accuracy by the step at which the alternative method exceeds that accuracy. Speedup is 0 if it never achieves a greater max accuracy (marked by no-fill points).

## 5 CONCLUSION

In this work, we introduced Gradient Informed Selection Training (GIST), a method for active data selection that leverages a small holdout set to guide training via gradient alignment. We demonstrated that aligning per-example gradients from each training superbatch with the holdout set gradient leads to strong selection of training samples, outperforming baselines such as RHO-LOSS, Coherence, and Random selection. We find that this method is not only able to achieve large speedups, but able to significantly increase the final model performance. We demonstrate the robustness of our model in both the text and image domain on three datasets. We conducted ablations of selection fraction hyperparameter and found that the GIST method is robust to a wide selection of values. We conducted an ablation to demonstrate the importance to the performance of this method of having a separate holdout set which is not trained on directly. In addition, we conducted analysis of which examples and which classes are selected during training on the Clothing-1M dataset and found many interesting trends that point toward the importance of running the data selection continuously during training as which data is important to the model changes greatly as it converges. Our method shows strong results on both noisy datasets like Clothing1M and does so without requiring teacher models, auxiliary objectives or heuristics.

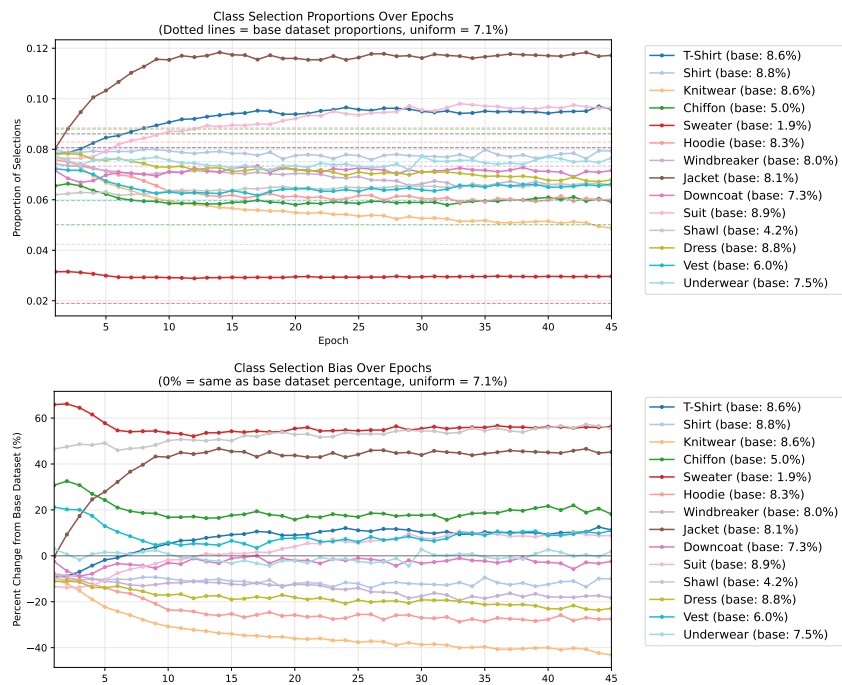

Figure 9: **Top:** Effective proportion of the selected train set for each class for the Clothing-1M dataset during GIST training with 60% selection ratio for each epoch of training. Dotted line for each class corresponds to the class proportion in the dataset. **Bottom:** Effective over/under-sampling rate of each class compared to it's base dataset proportion.

## 6 DISCUSSION AND FUTURE WORK

One promising application of GIST is learning in privacy-sensitive settings. In scenarios where a confidential or copyrighted dataset cannot be used directly for training (due to legal, ethical, or security constraints) GIST enables indirect training by selecting subbatches of general training data that approximate the gradient directions of the confidential set. Since the model never directly optimizes over the private data nor stores any of its examples or labels, this approach could satisfy certain compliance requirements while enabling more generalizable learning. Potential applications include medical diagnosis, governmental applications, and federated learning, where privacy-preserving training is crucial.

Another avenue for future research is improving the quality of the holdout set. While we currently use a small random subset, future work could explore construction of the holdout set using dataset distillation or optimized processes. Additionally, adaptive holdout sets that evolve during training may allow GIST to better capture shifting model uncertainties over time.

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

# A  APPENDIX

## A.1  CLOTHING-1M EXPERIMENT DETAILS

All models are pretrained on ImageNet, following the setup in RHO-LOSS Mindermann et al. (2022), and then fine-tuned on Clothing-1M for a total of 6 effective training epochs. We define one effective epoch as processing a number of samples equal to the full dataset size. For example, with a subset selection fraction of 0.10, each training pass sees (updates weights on) only 10% of the data, so 60 such passes are needed to match the sample count of 6 full epochs. For the holdout set, we randomly split out 200 images per class (2800 total), which is 0.28% of the entire training set size.

## A.2 MODELS

We use the ViT-Small architecture Dosovitskiy et al. (2021) for all experiments, as it offers a balance between performance and efficiency. Moreover, unlike architectures that include BatchNorm layers Ioffe & Szegedy (2015) (e.g., ResNets He et al. (2015)), ViTs enable gradient averaging without batch-dependent side effects, which is important for our selection strategy. For the RHO-LOSS experiments, we use a smaller ViT-Tiny model as the teacher.

## A.3 HYPERPARAMETERS

All models are trained using the AdamW optimizer with $\beta_1 = 0.9$, $\beta_2 = 0.999$, weight decay $= 0.05$, eps $=$ 1e-8, learning rate $= 0.001$. For all experiments, we use a batch size of 320, following the RHO-LOSS setup for reproducibility. We finetuned all weights, not just the final layer.

## A.4 CLOTHING-1M: CONSISTENTLY SELECTED IMAGES

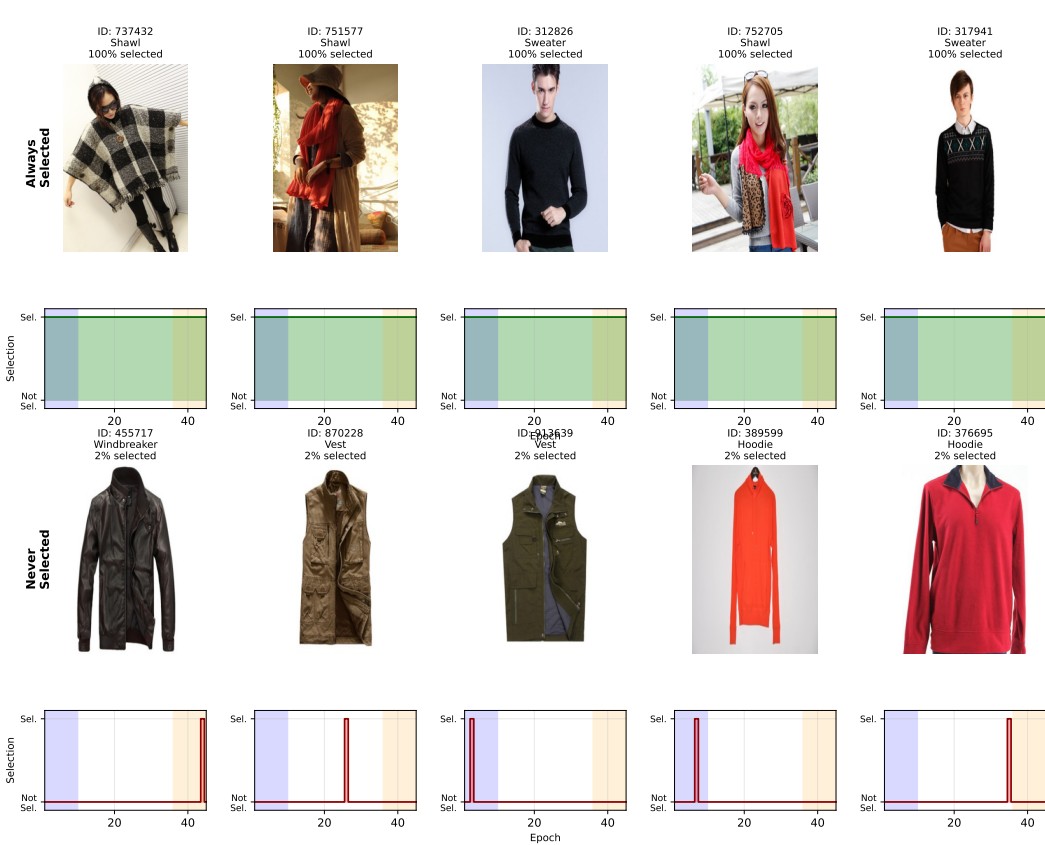

## A.5    CLOTHING-1M: LATE AND EARLY BLOOMERS

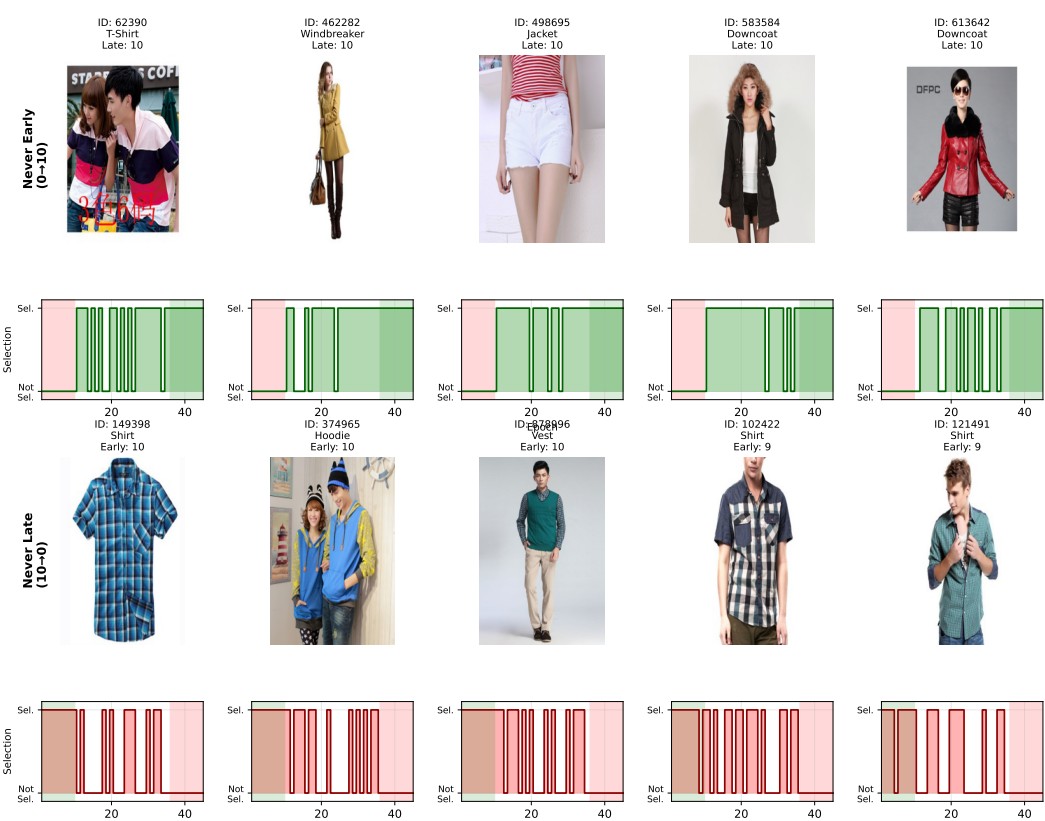