# OpenReview forum: "Look but Don’t Touch: Gradient Informed Selection Training"
_ICLR.cc/2026/Conference — Submitted to ICLR 2026_

### Official Review · Reviewer_xGis · 2025-10-17

**Soundness:** 2
**Presentation:** 1
**Contribution:** 2
**Rating:** 2
**Confidence:** 2

**Summary:**

The paper proposes **GIST**, which uses a small, class-balanced holdout set drawn from the training distribution to compute an average (last-layer) gradient;  within each superbatch, it selects examples whose gradients have the highest cosine alignment with this holdout gradient and updates only on that sub-batch.

**Strengths:**

Simple, single-network selection rule, easy to implement and reason about.

**Weaknesses:**

1.  The presentation appears rushed.  On  Page 4 and Page 7, oversized figures occupy almost the entire page, wasting space.  This could be presented in a more reasonable way.
2.  The abstract’s claim — “The amount of data available for training foundation models is far greater than our amount of compute. ” — seems at odds with current LLM perspectives, where the amount of available data is also crucial.  I suggest adding reasonable qualifications to this statement.
3.  Key design decisions (holdout construction, class balance, refresh schedule) are buried in the appendix and should be surfaced in §3.
4.  The experiments seem to miss some important baselines, such as training on the full dataset, training only on the holdout set, and strong baselines like InfoBatch.
5.  Some minor wording issues, e.g., in Figure 5 the term “Vanilla” is not explained — is it the random method?
6.  The GIST method description is too brief, making it hard to clearly understand the motivation and the plausible source of effectiveness.  Even a linearized analysis (last-layer surrogate) showing why alignment with a proxy of test-risk gradient can reduce generalization error would help.

**Questions:**

1.  I believe the holdout set has a large impact on the final results, and the method aligns gradients to this holdout set.  Doesn’t this essentially amount to selecting a better subset from the large dataset for training?  Moreover, according to the authors, this subset seems to be class-balanced in A.1 (“we randomly split out 200 images per class”).  So are the gains mainly coming from this more balanced dataset?  You could try training directly on this balanced holdout set to see the results.  The current results do not seem to support GIST method’s effectiveness very well.
2.  In Figure 8, when the selection ratio is 1, different methods produce the same results.  In the experiments, are the RHO-LOSS holdout set and the GIST holdout set the same?  Why would random, which has no holdout set, achieve the same performance?

---

### Official Review · Reviewer_GEkM · 2025-10-24

**Soundness:** 1
**Presentation:** 1
**Contribution:** 1
**Rating:** 0
**Confidence:** 3

**Summary:**

Similar to RHO-loss, this work uses a holdout set (and potentially a reference model) to perform prioritized training, but instead of focusing on a loss-based approach, it uses a gradient alignment. I have a number of major concerns with this paper, as it stands.

**Strengths:**

- interesting topic area (prioritized training), and if the claim is true (improvement over rho-loss), that would be notable. However, I am not convinced that this is the case.

**Weaknesses:**

- **baseline differences**: in Fig 1, RHO-LOSS does not outperform uniform selection. However, in Mindermann et al 2022, RHO-LOSS substantially outperforms uniform selection, and achieves ~>71% accuracy for resnet-50 (and other achitetures). this could be an architecture difference, but it begs the questions about the reason behind the achitecture being used in Fig 1 (which isn't stated), and makes me skeptical of the results. This is a major problem for the paper—without a correctly implemented baseline, I am not sure about whether the method truly improves results. This needs to be understood and explained.
- I am unsure about whether the approach truly makes sense. For example, with clothing-1M, if the holdout set has noisy labels and with lots of redundancy, we explicitly do __not__ want to prioritize alignment to this gradient (because this gradient is bad)
- writing quality and polish, for examples, citations are in plain text (should be \citet or \citep). There is a whole page of figures on page 4, which should be avoided (and the figures are huge)

**Questions:**

- abstract states "3x faster wall clock time, using 6x fewer steps, and
  achieves 4% higher final accuracy than RHO-LOSS and uniform data selection", is this compared to RHO-LOSS or uniform data selection? They are different methods? Later, it seems these methods perform similarly, which seems to indicate something has not been implemented correctly.
- "Yet, current strategies fall short of the performance upper bound achievable if we could always train on the most beneficial examples." what is the evidence for this claim? do you have evidence for it?

---

### Official Review · Reviewer_ndQv · 2025-10-31

**Soundness:** 2
**Presentation:** 3
**Contribution:** 2
**Rating:** 2
**Confidence:** 5

**Summary:**

This paper proposes an algorithm called “Gradient Informed Selection Training” (GIST) which the authors claim to speed up training while preserving or even reducing test set performance. The intuition is that models only need to be trained on minibatches of data whose gradients are aligned with the average gradient over the whole dataset. Therefore, we can adaptively select subsets of the dataset to train on, without sacrificing quality.

To start, the algorithm holds out a fraction of the training set (in the experiments on the Clothing-1M dataset, they hold out 0.28% of the training examples). Then, in each “superbatch” sampled from the training set, identify a fraction (e.g., 60%) of examples whose last-layer gradients have the highest cosine similarity with the holdout set’s last-layer gradients. Only backprop on these selected examples.

The authors conduct experiments on the Clothing-1M, WebVision, and OpenWebText2 datasets. Ablation studies show consistent training speed ups with no drop in test set performance for selection ratios greater than ~0.3.

**Strengths:**

**S1) Intuitive idea with adaptive data selection**

The idea of adaptively selecting training data on every training step is intuitive and straightforward. The algorithm is easy to understand. The proposed adaptive approach is notably different from the coreset selection problem which identifies a fixed coreset for all epochs of training.

**S2) Strong empirical performance**

The empirical metrics shown are strong, demonstrating significant speedups in both wall-clock time and # of training steps.

**Weaknesses:**

**W1) Reproducibility challenges**

As this paper is an empirical paper, reproducibility of the results is critical. There is no code provided either via the supplementary materials or via an anonymous repository link. Furthermore, there are not enough details provided in the paper nor appendix about the model architectures used. For example, Appendix A.2 only discusses the models used for the Clothing-1M dataset, but not the WebVision and OpenWebText2 datasets.

**W2) Lack of error bars**

None of the plots presented in the paper show any error bars, and therefore it is hard to understand whether the results are genuine or statistical flukes. Notably, Figure 7 is rather counter-intuitive, with non-monotonic behavior of the validation loss as a function of the amount of training data selected. It would be especially useful to see error bars (e.g., mean +/- standard error) on Figures 7-8 over at least 3 independent runs.

**W3) Insufficient justification for claim that GIST is useful in datasets with noisy labels**

The authors claim that GIST achieves “acceleration in training for large-scale noisy datasets,” but the claim is not thoroughly investigated. To really prove this point about the benefit of GIST on *noisy* datasets, the authors should consider experiments where noise is systematically added to the dataset labels. Ideally, we would see that the performance gain of GIST over baselines increases as a function of label noise.

**W4) Formatting errors**

These errors do not affect my score, but need to be corrected nonetheless:

- formatting of in-text citations throughout the paper. For example, in the 2nd line of the introduction, “Foundation models such as CLIP Radford et al. (2021) and GPT Brown et al. (2020)” should be instead written as “Foundation models such as CLIP (Radford et al., 2021) and GPT (Brown et al., 2020)”

- Line 299 and onwards: opening quotation marks should be different from closing quotation marks

- Line 307 and the rest of that paragraph: “likelyhoods” should be spelled “likelihoods”

**Questions:**

**Q1) How does GIST compare against coreset approaches?**

In the Related Work section, the authors mention data subset selection methods such as Coleman et al. (2020). How does GIST compare against these coreset approaches? Intuitively, it seems like the coreset approaches could be faster, because the data selection only occurs once, as opposed to adaptively.

**Q2) Why does RHO-LOSS perform worse at larger selection fractions?**

Figure 8 suggests that the RHO-LOSS baseline seems to perform on-par or worse than random when the selection ratio is larger than 0.3. Why might that be the case? And why does GIST overcome that problem?

**Q3) Why is SNR lower in vision than text data?**

In the introduction, the authors write, “we focus on vision model training, where the signal-to-noise ratio in data is often lower than in text or structured domains.” This is missing an explanation or a citation.

---

### Official Review · Reviewer_ufiG · 2025-11-01

**Soundness:** 3
**Presentation:** 3
**Contribution:** 3
**Rating:** 6
**Confidence:** 4

**Summary:**

This paper presents **GIST (Gradient Informed Selection Training)**, an active data selection method that, at each step, computes per-example (last-layer) gradients for a large **superbatch** and selects a smaller **subbatch** whose gradients are most aligned (via cosine similarity) with the gradient computed on a small, fixed **holdout set** drawn from the training distribution. The model is updated only on the selected subbatch, steering learning toward directions that generalize. This paper evaluates GIST on noisy vision datasets (**Clothing-1M**, **WebVision**) and text (**OpenWebText2**), reporting **2–6× fewer steps** and **≈3× wall-clock speedups** to reach a target performance, along with **higher final accuracy** than baselines including RHO-LOSS, coherence selection, and random selection. Ablations over selection fraction and the use of last-layer gradients illuminate compute/accuracy trade-offs.

**Strengths:**

- **Originality.** This paper reframes data selection as **gradient alignment to a held-out proxy** for test behavior, avoiding teacher–student machinery and reducible-loss estimation while remaining simple and model-agnostic.
- **Quality.** The algorithm is clear and practically grounded (last-layer gradients to reduce cost; superbatch→subbatch selection). Results span image and text, consistently improving both convergence speed and final metrics; ablations cover selection fraction and gradient choices.
- **Clarity.** Figures and the step-by-step procedure make the intuition concrete (holdout-aligned directions vs. coherence to the superbatch). Experimental protocols are explicit.
- **Significance.** On noisy, large-scale data, this paper achieves notable speed/accuracy gains over strong selection baselines, suggesting broad utility for efficient training under compute constraints.

**Weaknesses:**

- **Compute overhead details.** While the method reports a step-time slowdown from per-example gradients, this paper does not provide a full profiling (FLOPs, memory, throughput) vs. superbatch size and selection fraction, making operational trade-offs harder to assess.
- **Holdout sensitivity.** Performance may depend on holdout composition (size, class balance, cleanliness). A systematic sensitivity study would clarify robustness, especially under distribution shift.
- **Baselines/architectures.** Beyond RHO-LOSS/coherence/random, comparisons to **BayesianSelection/GLISTER** and to additional backbones (e.g., ResNets/CLIP for vision; larger LMs) would strengthen generality claims.
- **Theory.** This paper lacks guarantees (e.g., when holdout-aligned updates provably aid generalization or avoid bias), leaving alignment-to-generalization links empirical.

**Questions:**

1. **Holdout design.** How sensitive are results to holdout size/quality/class balance? Can an **adaptive** holdout improve performance without inducing overfitting to its gradient direction?
2. **Compute profiling.** Please report **wall-clock throughput, memory footprint, and FLOPs** across (superbatch size, selection fraction), and discuss engineering tricks (vectorized per-example grads, caching, activation checkpointing).
3. **Broader baselines.** Include **Grad-Match/GLISTER** and stronger model families (ResNet/CLIP; larger GPT variants) to validate portability.
4. **Shift & robustness.** Evaluate when the holdout mismatches training/eval distributions; can curriculum the holdout (or reweight it) to handle drift?
5. **Ablate gradient choices.** Beyond last-layer, do block-wise or Fisher-weighted similarities help? Any efficient stochastic estimators that preserve gains with lower overhead?
6. **Privacy angle.** If used as a proxy for private data, what leakage risks remain and what guarantees (if any) can be claimed?

---

### Meta-Review · Area_Chair_KcHq · 2026-01-07

**Summary:**

No rebuttal was provided. The reviewers raise serious concerns, such as the reproducibility challenges, the lack of error bars, formatting errors, the proposed method make sense or not, poor formatting, missing important baselines, etc.

**Reviewer Concerns:**

No rebuttal was provided so no concerns were addressed.

**Reviewer Scores:**

No rebuttal was provided so I don't expect any changes to the reviewers' scores.

---

### Decision · Program_Chairs · 2026-01-26

Reject